# Methylglyoxal and Advanced Glycation End Products (AGEs): Targets for the Prevention and Treatment of Diabetes-Associated Bladder Dysfunction?

**DOI:** 10.3390/biomedicines12050939

**Published:** 2024-04-23

**Authors:** Akila Lara Oliveira, Mariana Gonçalves de Oliveira, Fabíola Zakia Mónica, Edson Antunes

**Affiliations:** Department of Translational Medicine, Pharmacology Area, Faculty of Medical Sciences, University of Campinas (UNICAMP), Campinas 13084-971, SP, Brazil; a192906@dac.unicamp.br (A.L.O.); marigo@unicamp.br (M.G.d.O.); fzm@unicamp.br (F.Z.M.)

**Keywords:** dicarbonyl stress, RAGE, oxidative stress, polyphenols, metformin, alagebrium

## Abstract

Methylglyoxal (MGO) is a highly reactive α-dicarbonyl compound formed endogenously from 3-carbon glycolytic intermediates. Methylglyoxal accumulated in plasma and urine of hyperglycemic and diabetic individuals acts as a potent peptide glycation molecule, giving rise to advanced glycation end products (AGEs) like arginine-derived hydroimidazolone (MG-H1) and carboxyethyl-lysine (CEL). Methylglyoxal-derived AGEs exert their effects mostly via activation of RAGE, a cell surface receptor that initiates multiple intracellular signaling pathways, favoring a pro-oxidant environment through NADPH oxidase activation and generation of high levels of reactive oxygen species (ROS). Diabetic bladder dysfunction is a bothersome urological complication in patients with poorly controlled diabetes mellitus and may comprise overactive bladder, urge incontinence, poor emptying, dribbling, incomplete emptying of the bladder, and urinary retention. Preclinical models of type 1 and type 2 diabetes have further confirmed the relationship between diabetes and voiding dysfunction. Interestingly, healthy mice supplemented with MGO for prolonged periods exhibit in vivo and in vitro bladder dysfunction, which is accompanied by increased AGE formation and RAGE expression, as well as by ROS overproduction in bladder tissues. Drugs reported to scavenge MGO and to inactivate AGEs like metformin, polyphenols, and alagebrium (ALT-711) have shown favorable outcomes on bladder dysfunction in diabetic obese leptin-deficient and MGO-exposed mice. Therefore, MGO, AGEs, and RAGE levels may be critically involved in the pathogenesis of bladder dysfunction in diabetic individuals. However, there are no clinical trials designed to test drugs that selectively inhibit the MGO–AGEs–RAGE signaling, aiming to reduce the manifestations of diabetes-associated bladder dysfunction. This review summarizes the current literature on the role of MGO–AGEs–RAGE–ROS axis in diabetes-associated bladder dysfunction. Drugs that directly inactivate MGO and ameliorate bladder dysfunction are also reviewed here.

## 1. Introduction

Methylglyoxal (MGO) is a highly reactive α-dicarbonyl compound endogenously generated during the glycolytic pathway [1]. Hyperglycemia in diabetic and obese patients markedly elevates the plasma and urine levels of MGO as a consequence of the glycolytic overload [2]. The abnormal accumulation of MGO has been referred to as dicarbonyl stress, which may be implicated in many diseases [3]. Methylglyoxal promotes post-translational modification of peptides or proteins, ultimately leading to the formation of advanced glycation end products (AGEs), the most studied of which include arginine-derived hydroimidazolone (MG-H1) and carboxyethyl-lysine (CEL) [1]. MGO also covalently modifies DNA, leading to nucleic acid AGE formation, consisting mainly of guanine adducts. AGEs bind their cell membrane-anchored ligand receptor, termed RAGE [4], triggering multiple intracellular signaling pathways, including the activation of NADPH oxidase that leads to increased production of reactive oxygen species (ROS), thus contributing to generate a pro-oxidant environment. Diabetic bladder dysfunction (DBD) is a highly prevalent condition that may affect the detrusor, nerve fiber terminals, urothelium, and urethra, and manifests as storage problems such as OAB and urge incontinence, and voiding problems such as decreased sensation and increased capacity [5,6]. DBD may progress from detrusor overactivity at initial stages to detrusor underactivity at advanced stages of this disease, a condition defined by the International Continence Society (ICS) as contraction of reduced strength and/or duration, resulting in prolonged bladder emptying and/or a failure to achieve complete bladder emptying within a normal timespan [5,7,8,9]. The underactive bladder comprises mostly voiding phase symptoms such as slow stream, intermittency, hesitancy, feeling of incomplete emptying of the bladder, and urinary retention [10]. Increased capacity and decreased sensation, together with recurrent urinary tract infections, may also be present in DBD [11,12]. Preclinical models of type 1 (streptozotocin, Akita mice) and type 2 diabetes (high-fat diets, ob/ob and db/db mice) have provided further evidence confirming the relationship between diabetes and obesity with voiding dysfunction. However, little is known about the importance of MGO generation and, hence, AGEs–RAGE activation in the pathophysiology of diabetic-associated bladder dysfunction [13]. Interestingly, in mice treated orally with MGO for prolonged periods, voiding spot assays in conscious mice and urodynamic evaluation in anesthetized mice revealed significant increases in total void volume, volume per void, micturition frequency, and nonvoiding contractions number, along with enhanced in vitro bladder contractility [14]. In addition, elevated levels of MGO, AGEs, RAGE, and ROS were found in bladder tissues from mice chronically treated with MGO, pointing out that they could be important markers of DBD pathophysiology [15]. Similar data were obtained in bladder tissues of diabetic obese ob/ob mice [16]. The antihyperglycemic drug metformin [17,18,19] and polyphenols like resveratrol and epigallocatechin-3-gallate [20] can directly scavenge MGO, explaining, at least in part, their capacity to ameliorate diabetes-associated bladder dysfunction. However, no clinical trials exist aiming to test inhibitors of the MGO–AGEs–RAGE signaling as potential drugs to prevent and treat manifestations of diabetes-associated bladder dysfunction. Therefore, the design and development of new drugs that inhibit the MGO–AGEs–RAGE axis may become an interesting approach for the prevention and treatment of bladder dysfunction in diabetic conditions. The present review aimed to provide an updated overview on bladder dysfunction in diabetic and obesity conditions in animals and humans, emphasizing the MGO–AGEs–RAGE signaling pathway as a potential mechanism implicated in the pathophysiology of this disorder, focusing on bladder overactivity. Drugs that inactivate MGO or inhibit AGEs formation in parallel to reducing diabetic-associated bladder dysfunction are also reviewed here. 

## 2. Lower Urinary Tract Symptoms (LUTS) and Overactive Bladder (OAB) Syndrome

Urinary bladder function is regulated by a complex interaction of efferent and afferent fibers from the autonomic nervous system and somatic innervation [21]. An imbalance between these systems leads to lower urinary tract symptoms (LUTS), which comprise storage, voiding, and post-micturition symptoms [5]. Storage symptoms consist of altered bladder sensation, increased daytime frequency, nocturia, and urgency incontinence, whereas voiding symptoms consists of hesitancy, intermittency, weak or irregular stream, straining, and terminal dribble. Post-micturition symptoms include dribbling and sensation of incomplete voiding. The storage symptoms are generally more bothersome than voiding or post-micturition symptoms, as observed in both men and women. Overactive bladder (OAB) syndrome is a subgroup of storage symptoms consisting mainly of urinary urgency. In men, LUTS typically occur in association with bladder outlet obstruction (BOO) secondary to benign prostatic hyperplasia (BPH), despite that it may occur independently of BOO or prostatic diseases, whereas in women, the most frequent LUTS is stress urinary incontinence [22]. Epidemiological studies have shown OAB to be a widely prevalent condition in men and women [5,23], with an incidence of 16.6% in a sample from Europe [24], 16.9% in women and 16.0% in men in a sample from the USA [25], and an overall prevalence of OAB of 18.9% in a South American population [26,27]. Epidemiological studies applying the ICS definition of OAB across multiple countries found a prevalence of 11–13% [5]. LUTS negatively impacts the social quality of life and sexual health of patients [28,29].

## 3. Association of Metabolic Syndrome and Diabetes with Urinary Bladder Dysfunction

Metabolic syndrome, the medical term for a combination of cardiometabolic risk factors such as central obesity, hyperglycemia, hypertension, and dyslipidemia, is critically involved in the onset of many cardiovascular diseases, being the leading cause of death worldwide [30]. Outside the cardiovascular system, metabolic syndrome associated with increased body mass index (BMI) represents an important risk factor for LUTS/OAB and urinary incontinence [31,32,33,34,35,36,37,38,39,40,41,42,43,44], despite some studies showing no positive association between metabolic syndrome and LUTS in men and women [45,46]. An association between metabolic syndrome and interstitial cystitis/bladder pain syndrome (IC/BPS) has been reported in women [47]. Elevated body mass index and diabetes also increase the risk of urinary tract infections and pyelonephritis [48,49]. Metabolic syndrome is also associated with LUTS secondary to benign prostatic hyperplasia [50,51,52]. Surgical and nonsurgical weight loss leads to improvements in stress urinary incontinence [53], even though a definite conclusion has not been achieved [54].

## 4. Bladder Dysfunction in Type 1 and Type 2 Diabetes in Patients and Animals

Diabetes mellitus is a chronic metabolic disease characterized by high blood sugar levels (hyperglycemia) as a result of abnormal insulin production and/or insulin function. The most common and bothersome urological complication of diabetes mellitus is DBD (or diabetic cystopathy), which affects more than 80% of individuals diagnosed with diabetes [55,56,57,58,59]. The pathophysiology of DBD is multifactorial and may involve alterations at all levels of the urinary tract, including the detrusor, urethra, urothelium, and innervation [60]. Clinical DBD manifestations consist of storage bladder problems such as OAB and urge incontinence, and voiding problems like poor emptying with resultant elevated post-void residual urine [7,12]. Increased capacity and decreased sensation together with recurrent urinary tract infections may also be present in DBD [11]. Preclinical models of type 1 (T1DM) and type 2 diabetes (T2DM) have provided further evidence confirming the relationship between metabolic diseases and bladder dysfunction [61,62,63,64,65,66,67]. 

T1DM can be mimicked by injection of streptozotocin (STZ) in rodents, a cytotoxic glucose analogue that destroys pancreatic β-cells due to its high affinity for glucose transporter 2 (GLUT2) [68,69]. Analysis of bladders from STZ-induced diabetes in male and female rodents (rats and mice) revealed increased bladder mass [70,71,72,73,74,75], which is suggested to represent a physical adaptation to increased urine production [76,77,78]. Despite that high glucose levels and diabetic polyuria have been proposed as pathophysiological mechanisms explaining bladder enlargement in the STZ model [79], a recent study comprising different models of diabetes in rodents, including T1DM, did not confirm such a proposition [80]. Insulin administration can prevent, or even reverse, most of the morphological, functional, and molecular bladder alterations in the STZ model [79,81,82,83]. Moreover, increases in both volume and frequency of micturition [70,73], as well as in urinary frequency, capacity, and number of nonvoiding contraction (NVCs) [66,84], have been reported in STZ-induced diabetes, as revealed by urodynamic studies. Spontaneous voiding spot assays [85,86] also revealed significant increases in voiding frequency, total voided volume, and mean volume per micturition in STZ-injected mice [87], which are paralleled by in vitro detrusor overactivity [66,88]. However, after prolonged hyperglycemia and insulin resistance in response to STZ, bladders may progress to an underactive detrusor and an inability to produce an effective voiding [64] through mechanisms mediated by the activation of NLRP-3 inflammasome [89]. Therefore, in STZ-induced diabetes, a temporal effect of diabetes on bladder activity has been established, that is, an early phase of compensatory followed by a later phase of decompensated bladder function [7,63,77,78]. In female Akita mice (T1DM model), diabetic bladder dysfunction also progresses from overactivity to underactivity [90]. At the molecular level, the impairment of the nitric oxide—soluble guanylate cyclase (sGC)—cyclic GMP signaling [82,91] and NLRP3 inflammasome activation in urothelial cells [89] have been proposed as a critical mechanism contributing to bladder dysfunction. Nevertheless, conflicting data on different parameters of bladder activity in animal models of STZ-induced diabetes models have been obtained, which may possibly rely on both animal species and strain used, in addition to the disease time course [77,78,92,93]. Experimental T1DM in rats and rabbits can also be induced using alloxan, a hydrophilic unstable compound that shares a structure similar to glucose [94]. Increases in bladder weight, detrusor smooth muscle cells, capacity, and urinary output, along with irregular bladder contractions, were observed in alloxan-induced diabetic rats [95,96,97,98]. In rabbits made diabetic by alloxan, an increase in bladder weight [99] and a reduction in the vitro bladder contractions to carbachol were reported [100].

The main T2DM models that result in hyperinsulinemia and insulin resistance rely on allowing animals free access to diets highly enriched in fats [101,102]. In addition to producing the classical obesity-associated vascular dysfunction, male mice fed high-fat diets progress to an overactive bladder phenotype, as evidenced mainly by filling cystometry in anesthetized and awake rats and mice [103,104,105]. The resulting increased body weight, hyperglycemia, and insulin resistance by prolonged high-fat diet intake in mice is also accompanied by in vitro bladder overactivity as a consequence of high extracellular calcium influx through L-type voltage-operated calcium [106,107,108,109]. The importance of calcium channels to bladder dysfunction has also been confirmed in diabetic db/db mice [110]. High-fat diet-fed obese mice also display impaired urethral smooth muscle relaxations [111,112] and prostate hypercontractility [105,113], which are suggested to contribute to the resulting bladder overactivity. Impaired striated urethral muscle contractions were reported in Zucker obese rats [114]. Contrasting to these studies, no evidence of bladder dysfunction was observed in obese mice fed a high-fat diet for 16 weeks, as assessed by void spot assays [115]. The temporal effects (up to 42 weeks) of different diets consisting of fructose, cholesterol, and lard, at varying proportions and combinations, on 24 h urinary behavior and conscious cystometry were investigated in rats [116]. Compared with the control group, the total voided volume was lower in all experimental diets, and animals receiving 32.5% lard diet alone exhibited decreases in bladder capacity, mean voided volume, and inter-micturition intervals that were indicative of an overactive bladder phenotype [116]. 

Leptin is a satiety hormone that is synthesized by adipocytes, the levels of which increase with the adipose tissue mass [117]. Mice genetically deficient in leptin (ob/ob) or in the leptin receptor (db/db) are hyperphagic, obese, hyperinsulinemic, and hyperglycemic, and have been widely used as a T2DM model [118]. Similarly, in the STZ- or diet-induced obesity models, ob/ob male mice exhibit bladder dysfunction characterized by increases in urine volume and in vitro bladder smooth muscle contractions [119]. Increases in total void volume and volume per void with no alterations of spot number were observed in five-week-old male and female ob/ob mice, as evaluated by void spot assays [16]. Four- and six-month-old ob/ob mice exhibited some degree of bladder dysfunction such as increases in total urine volume and number of primary void spots, although that depended on animal sex and animal age [115]. In male db/db mice, increases in bladder weight, voiding frequency, and capacity together with elevated in vitro contractions were described [110]. Increases in detrusor smooth muscle area, urothelium area, and collagen content were also reported in male and female db/db mice at 12, 24, or 52 weeks of age, which was suggested to reflect a compensatory response to the increased urine output [120]. Double-knockout hepatic-specific insulin receptor substrate 1 and 2 (IRS1 and IRS2) female mice that develop T2DM exhibit bladder overactivity, high frequency of nonvoiding contractions, decreased voided volume, and dispersed urine spot patterns [121]. 

## 5. Methylglyoxal–Advanced Glycation End Products (AGEs)–RAGE Signaling Pathway

The abnormal accumulation of highly reactive dicarbonyl compounds as a consequence of glycolytic overload has been referred to as dicarbonyl stress [1,3]. 1,2-Dicarbonyl compounds include glucosone, 3-deoxyglucosone, methylglyoxal (MGO), and glyoxal, but MGO is one of the most studied, given that it exerts a critical role in diabetes-associated cardiovascular complications, such as diabetic nephropathy, endothelial dysfunction, postinfarct remodeling, and impairment of insulin signaling [122,123,124,125,126]. Methylglyoxal, chemically referred as acetylformaldehyde, 2-ketopropionaldehyde, pyruvaldehyde, or 2-oxo-propanal, is a highly reactive dicarbonyl compound formed endogenously from 3-carbon glycolytic intermediates of glycolysis (dihydroxyacetone phosphate and glyceraldehyde-3-phosphate), although it can also be generated as a byproduct of protein, lipid, and ketones [127,128]. In addition to the endogenous production in mammalian cells, MGO may be present at marked levels in many food products and beverages, as well as in microorganisms [129]. In healthy conditions, glyoxalases (Glo) are the most important enzymatic detoxification system that converts MGO into its end-product D-lactate [1]. Glyoxalases comprise two major enzymes, namely, Glo1 (lactoylglutathione methylglyoxal lyase) and Glo2 (hydroxyacylglutathione hydrolase), with Glo1 described as a rate-limiting enzyme [130,131,132]. Interestingly, the increased levels of glucose and MGO are normalized in Glo-1 transgenic rats after induction of diabetes by intravenous injection of STZ [123,133].

The endogenous process by which endogenous MGO promotes post-translational modification of peptides or proteins, ultimately leading to generation of AGEs, is referred to as glycation [2]. The main MGO-derived AGEs in mammalian metabolism are arginine-derived hydroimidazolone (MG-H1) and carboxyethyl-lysine (CEL) [134], but other AGEs may be generated depending on the dicarbonyl species formed [135]. The mechanism of MG-H1 generation involves the replacement of the hydrophilic positively charged arginine residue by an uncharged, hydrophobic MG-H1 residue, producing misfolding and activation of the unfolded protein response [1,136]. Incubation of human plasma from heathy donors with different concentrations of MGO (10 and 100 µM) for 24 h induced a time- and dose-dependent increase in MG-H1 levels, as detected within the first 6 h [137].

Once generated, AGEs bind their cell membrane-anchored ligand receptor (termed RAGE), which is a member of the immunoglobulin superfamily of cell surface receptors able to recognize endogenous ligands [4]. Structurally, RAGE consists of three immunoglobulin domains, that is, (i) an extracellular part consisting of one V type and two C types (C1 and C2), (ii) a transmembrane spanning helix, and (iii) a short, highly charged intracellular cytoplasmic “C” terminal tail that is primarily associated with the downstream signaling pathways [138]. The extracellular domain devoid of cytoplasmic and transmembrane domains is called soluble RAGE (sRAGE), which comprise two forms, namely, cleaved RAGE (cRAGE) and endogenous secretory RAGE (esRAGE or RAGEv1) [139]. cRAGE is generated at the cell surface by the proteolytic cleavage of RAGE at the boundary between its extracellular and transmembrane portions, whereas esRAGE results from alternative splicing of RAGE pre-mRNA. In advanced chronic kidney disease (CKD) patients, an inverse association between risk of mortality and cRAGE/esRAGE ratio was reported [140]. RAGE plays an important role in the innate immune response and as a mediator of proinflammatory processes, triggering multiple intracellular signaling pathways, including the generation of proinflammatory mediators such as IL-1β, VCAM-1, and TNF-α via the transcription factor NF-κB [138] and phosphorylation of JNK and p38MAPK [141]. Furthermore, many of the RAGE actions have been attributed to the activation of NADPH oxidase [142], which leads to excess formation of ROS, thus contributing to generate a pro-oxidant environment [143,144,145]. Figure 1 illustrates the MGO–AGEs–RAGE signaling and glyoxalase system (Glo1 and Glo2 enzymes). 

Plasma levels of MGO are markedly elevated in conditions of hyperglycemia associated with diabetes mellitus in men and women, as usually detected by liquid chromatography–mass spectrometry, enzyme-linked immunosorbent, or electrochemical biosensor assays [146,147,148,149,150,151,152,153]. The patients included in these studies comprised men and women aged 54–61 years old, insulin- and noninsulin users, with accompanying diseases such as chronic renal failure, diabetic nephropathy, and coronary heart disease. Obese patients also have increased MGO levels in plasma, which can be even higher if the obese patients have diabetes [125,154]. The urine levels of MGO in diabetic patients are also higher than those in nondiabetic individuals [155]. In addition, in healthy volunteers, a rapid increase (49 min) in plasma levels of MGO was observed after oral glucose tolerance test (OGTT) [152]. Likewise, fasted healthy mice intraperitoneally injected with a glucose solution displayed a rapid elevation in plasma levels of MGO, as detected at 30 min after glucose administration [137]. Interestingly, lower levels of Glo1 activity in red blood cells paralleled the increased plasma MGO levels in T2DM patients displaying acute coronary syndrome [156]. 

## 6. MGO–AGEs–RAGE Axis as a Key Player of Bladder Dysfunction in Animals and Humans

There is a large amount of data on MGO in different organs at physiological and pathological conditions [2], but surprisingly few studies have explored the role of the MGO–AGEs–RAGE signaling pathway in the pathophysiology of the lower urinary tract system. The existing literature in this field has been restricted to bladder pain via the release of high mobility group box 1 protein (HMGB1) [157,158] and bladder cancer [159,160], which are not the focus of the present review.

In T2DM patients diagnosed with moderate/severe LUTS, serum levels of AGEs are positively correlated with symptoms and overactive bladder, suggesting that levels of AGEs may be early markers of diabetes-associated LUTS [13]. In addition, an immunohistochemical study in human bladders showed positive sites for carboxymethyl-lysine and pentosidine in the connective tissue between muscle bundles and muscle fibers, suggesting that extracellular matrix is the main site of action for AGE accumulation [161]. The MG-H1 free adduct has been described as the most responsive AGE associated with chronic kidney disease status, with higher levels in diabetic compared with nondiabetic individuals [162]. The MG-H1 residue contents of plasma protein are also elevated in male spontaneously diabetic Torii (SDT) rats at the age of 16 weeks [163].

The model of chronic overload intake of MGO at doses of 50 to 75 mg/kg for 6 to 12 weeks, as supplemented in the drinking water of the animals or injected intraperitoneally in rats and mice, has been shown to mimic some cardiovascular complications of diabetes in the absence of hyperglycemia such as endothelial dysfunction, microvascular damage, atherogenesis [164,165,166,167], cardiac dysfunction [168,169], and renal damage [170,171,172] (Table 1). However, the direct contribution of MGO to bladder dysfunction remains poorly investigated. This model of exogenous animal supplementation with MGO clearly differs from the classical diabetic animals in that MGO is not generated from the endogenous glucose metabolism, and, consequently, does not itself affect the glucose levels and insulin sensitivity [164,173,174,175]. Intake of MGO to healthy mice for 7 days (500 to 2000 mg/kg) significantly increased the urine levels of this dicarbonyl molecule [176]. Serum levels achieved by a 12-week intake of 0.5% MGO to healthy mice [174,175,177] reached comparable levels to those found in plasma of diabetic/obese individuals [125,154]. Levels of MGO levels were also increased in both plasma and urine after a 6-month MGO administration to mice at the doses of 200 mg/kg [176] and 500 mg/kg [178]. Likewise, in high-fat fed mice, levels of plasma and urine levels of MGO were significantly higher than animals kept on low-fat diet [179]. Diabetic obese ob/ob mice also displayed elevated serum MGO compared with normoglycemic animals [16] (Table 1).

Bladders from male mice treated orally with MGO for 4 weeks revealed tissue disorganization, partial loss of the urothelium, and mucosal edema along with marked cell infiltration [14]. Urodynamic evaluation (cystometric assays) in these male animals showed marked increases in micturition frequency and number of nonvoiding contractions (NVCs) with no alterations in bladder capacity [14]. Cystometric assays in male mice treated orally with MGO for an extended period (12 weeks) showed significant increases in the frequency of NVCs, bladder capacity, inter-micturition pressure, and residual volume [177]. In female mice treated with MGO for 12 weeks, cystometric assays confirmed urodynamic alterations such as increases in NVCs frequency, bladder capacity, inter-micturition pressure, and residual volume [15]. Using the model of spontaneous void spot assay (VSA) on filter paper, male mice treated with MGO for 12 weeks revealed an increased volume per void with no changes in the spot number as compared with the untreated group. In the female group, this treatment increased the spot number (mainly the number of microvolume spots) but, rather, reduced the volume per void [177,180]. During the MGO treatment, no alterations in the water consumption are observed in any group [177]. In this VSA assay, the term thigmotaxis refers to the wall-seeking behavior, that is, the tendency of mice to urinate next to the walls of the cage, which is interpreted as a rational response related to the fear of predation [181,182]. In healthy conditions, mice of both sexes will urinate at the corner of the filter paper, and when the animal loses this outlet control, urinating in the center of the filter, this may indicate bladder dysfunction. In the 12-week MGO treatment, whist the male mice had 95% of the voided spots in the corners of the filter paper, the voided spots in the female group were also detected in the center of the filter, indicating an altered outlet behavior in favor of an overactive bladder phenotype. The in vitro contractions to electrical-field stimulation (EFS; neurogenic contractions) as well as those induced by selective muscarinic and purinergic P2X1 receptor activation (using carbachol and α, β-methylene ATP as receptor agonists, respectively) were also evaluated in bladders of male and female mice treated with MGO for 12 weeks (Table 2). In intact bladder preparations of male mice, higher contractions to EFS, carbachol, and α,β-methylene ATP were observed after MGO treatment [177]. In the female group, higher contractile responses to EFS and α,β-methylene ATP (but not to carbachol) were also observed in intact bladder preparations from animals treated with MGO for 12 weeks [15]. An increased carbachol-induce response by MGO treatment in the female mice is solely observed when the urothelium is removed from the preparations. Moreover, the higher EFS-induced contractions in the MGO group were normalized by prior tissue incubation with the selective TRPA1 blocker HC-030031, suggesting that MGO exposure via TRPA1 activation leads to enhancement of purinergic over cholinergic neurotransmission in the bladder [180] (Figure 2). Table 2 summarizes the main in vivo and in vitro bladder alterations observed in male and female mice treated with MGO for 12 weeks. 

In MGO-treated mice, elevated levels of AGEs and RAGE in bladder tissues were also observed [15,177]. Likewise, hyperglycemic diabetic leptin-deficient male and female mice (ob/ob) exhibit bladder dysfunction, as evidenced by the increases in total void volume and volume per void (void spot assay) in addition to high collagen content in the bladders [16]. These bladder alterations were associated with high levels of total AGEs, MG-H1 and RAGE found in bladder tissues, which is consistent with the findings that the AGE breaker alagebrium (ALT-711) at 1 mg/kg during 8 weeks in the drinking water nearly reversed all the molecular and functional alterations in ob/ob mice [16] (Table 3). 

## 7. Drugs Presenting Potential to Downregulate AGEs Formation and Oxidative Stress in Bladder Tissues

It is well established that NADPH oxidase and increased levels of superoxide anion (O_2_^−^) and hydrogen peroxide (H_2_O_2_) play a critical role in diabetic complications [55,183,184,185,186,187,188,189,190,191,192]. Oxidative stress at excessive levels also plays an important role in pathophysiology of bladder outlet obstruction [193], cyclophosphamide-induced cystitis [194], benign prostatic hyperplasia [195] and STZ-induced bladder dysfunction [196]. Hydrogen peroxide (H_2_O_2_) is reported to activate bladder afferent signaling inducing detrusor overactivity [197]. In human and dog bladders in vitro, H_2_O_2_ itself produced contractions and potentiated the contractions induced by electrical-field stimulation, an effect attenuated by the natural NADPH oxidase inhibitor apocynin [198]. Given obesity-associated bladder dysfunction correlates with increased oxidative stress and that MGO treatment leads to excess ROS production, it is plausible that drugs that inactivate MGO [20,199] or that protect bladder cells from the oxidative insult [200,201] offer an interesting approach to reduce the deleterious effects of AGEs in the bladder (Figure 3). Therefore, we summarized below some drugs reported to ameliorate bladder dysfunction in animals including some polyphenols and metformin whose protective mechanisms may be related to their ability to downregulate AGEs formation and oxidative stress in bladder tissues. 

### 7.1. Polyphenols: Resveratrol and Epigallocatechin-3-Gallate

Resveratrol is a polyphenol present in numerous plant-based foods that increases lipolysis and reduces lipogenesis in adipocytes, being suggested as a therapeutic alternative to treat obesity-related diseases [202,203]. Two-week therapy with resveratrol (100 mg/kg/day, given by gavage) in high-fat-diet-fed obese mice reduced the in vivo urodynamic changes, the in vitro bladder overactivity, and the ROS production in bladder tissues [104]. Resveratrol treatment also increased the nitric oxide levels and restored the impaired urethral relaxations in obese mice, an effect mimicked by the antioxidant enzyme SOD [112]. Likewise, the in vitro urethral hyperactivity was restored by resveratrol in obese mice [112]. In the bladders of STZ-induced diabetic rats, daily oral treatment with resveratrol (10 mg/kg) reduced the histological abnormalities and inhibited the expression and localization of markers of oxidative stress and DNA oxidative damage [204]. Intragastric administration of resveratrol (20 mg/kg/day) reduced bladder hypertrophy, tissue damage, inflammatory cell infiltration, and levels of inflammatory cytokines in the bladders of STZ-induced diabetic rats [205]. In the chronic prostatitis model in rats, oral administration of resveratrol (10 mg/kg) for 10 days reduced the resulting overactive bladder and fibrosis by reducing the protein expressions SCF, c-Kit, and p-AKT [206,207]. At the molecular level, resveratrol exhibited a high inhibition rate on the fluorescent formation of AGEs mainly due to scavenging free radicals and capturing MGO [20]. Epigallocatechin-3-gallate is another polyphenol compound present in green tea that has also favorable effects on bladder overactivity, as evidenced in ovariectomized rats fed standard chow [201] and high-fat, high-sugar diet [208]. Treatment with epigallocatechin-3-gallate reduced the expressions of transforming growth factor-β (TGF-β) and type I collagen, as well as the apoptosis and oxidative stress in the bladders [208,209]. In a bladder outlet obstruction (BOO) model in rats, intraperitoneal injection of epigallocatechin-3-gallate (4.5 mg/kg/day) reduced the histologic changes and submucosal endoplasmic reticulum (ER) stress-related apoptosis, recovering the bladder compliance and inter-contractile intervals [210]. At 6 mM, epigallocatechin-3-gallate was also shown to exert anti-AGEs activity through its capacity to strongly trap and inactivate MGO [211]. In diabetic db/db mice, 16-week oral administration of (+)-catechin (15, 30, and 60 mg/kg) directly trapped MGO, hence downregulating the downstream signal transduction and inflammatory response induced by AGE–RAGE interaction in the kidney [212]. Therefore, although the mechanisms behind the uroprotective actions of polyphenols on diabetes-associated bladder dysfunction deserve further investigation, they could involve the capacity of these molecules to directly trap MGO, further inhibiting MGO-induced glycation, AGEs formation, and RAGE activation.

### 7.2. Metformin

Metformin is a first-line pharmacological treatment for T2DM patients as monotherapy or in combination with sulfonylureas or dipeptidyl peptidase 4 inhibitors [213]. The orally administered doses of metformin (as immediate-release or extended-release formulations) usually vary from 0.5 to 2.5 g daily, being safety and effective for long-term glycemic control. Metformin is associated with low risk of hypoglycemia and documented cardiovascular benefits [214]. Metformin increases tissue sensitivity to insulin and decreases the levels of glycated hemoglobin by mechanisms involving the activation of adenosine monophosphate-activated protein kinase (AMPK) and non-AMPK pathways [215], but its exact mechanism of action remains largely incomplete [213]. Recently, metformin was shown to increase intestinal glucose uptake, influencing hepatic glucose production through a gut–liver crosstalk [216]. Metformin is among the molecules reported to strongly react with MGO [17], forming an imidazolinone metabolite [19]. In addition, in the plasma of T2DM patients, metformin, through its guanidine group, was shown to bind to MGO, reducing this dicarbonyl concentration [18], hence reducing AGEs formation, which paralleled a significant increase in Glo1 activity [217]. A two-week treatment of high-fat-diet-fed mice with metformin (300 mg/kg) reversed the bladder overactivity, as evidenced by in vivo and in vitro studies [106]. Metformin also normalized the enhanced serum levels of MGO and fluorescent AGEs in mice treated chronically with MGO [177]. In bladders of MGO-treated mice, metformin treatment reduced Glo1 expression and activity, urothelium thickness, and collagen content, as well as the in vitro and in vivo micturition dysfunction [177]. It is, therefore, plausible to suggest that the beneficial effects of metformin in obesity-associated bladder dysfunction rely at least in part on its MGO capturing property. Of interest, oral administration of metformin (150 mg/kg, gavage) reduced both bladder remodeling and dysfunction in models of partial bladder outlet obstruction in rats [218], erectile dysfunction in obese mice [219], and diabetic nephropathy in STZ-induced diabetes [220]. 

## 8. Concluding Remarks and Future Therapeutics

Diabetic bladder dysfunction is a highly prevalent condition manifesting as storage (such as OAB and urge incontinence) and voiding problems (poor emptying with resultant elevated capacity). Increased capacity and decreased sensation together with recurrent urinary tract infections may also be present in DBD. Preclinical models of T1DM and T2DM in rodents have provided further evidence confirming the relationship between diabetes and bladder dysfunction. Hyperglycemia in diabetic/obese patients significantly elevates the levels of α-dicarbonyl compounds, including MGO, in plasma and urine as a consequence of the glycolytic overload. MGO promotes post-translational modification of peptides and proteins, ultimately leading to the formation of AGEs such as MG-H1. AGEs bind their cell membrane-anchored ligand receptor RAGE, triggering multiple intracellular signaling pathways, among which ROS production at excessive levels plays a critical role. However, little is known about the importance of MGO generation and AGEs–RAGE activation in the pathophysiology of diabetic-associated bladder dysfunction. Voiding spot assays and cystometrical evaluation in mice treated chronically with MGO have revealed significant increases in total void volume, volume per void, micturition frequency, and nonvoiding contractions number, along with enhanced in vitro bladder contractility. Moreover, levels of MGO, AGEs, RAGE, and ROS are all elevated in the bladder tissues obtained from MGO-treated animals and diabetic ob/ob mice. The antihyperglycemic drug metformin and the polyphenols resveratrol and epigallocatechin-3-gallate can directly scavenger MGO, exerting uroprotective actions. Therefore, we propose here that evaluation of MGO, AGEs, and RAGE levels may constitute important biomarkers of DBD pathophysiology. The design and development of new drugs that inhibit the MGO–AGEs–RAGE axis may become an interesting approach for the prevention and treatment of bladder dysfunction in diabetic conditions.

## Figures and Tables

**Figure 1 biomedicines-12-00939-f001:**
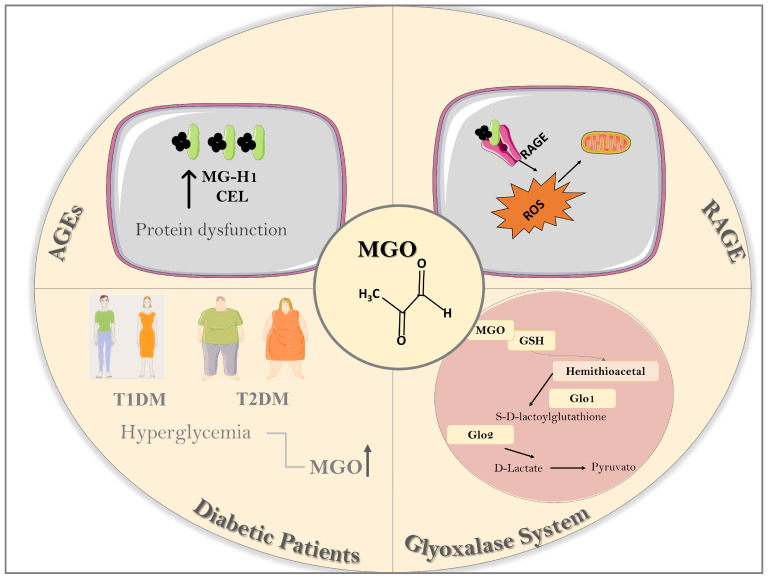
Methylglyoxal (MGO) is a highly reactive α-dicarbonyl compound generated endogenously during the glycolytic pathway. Hyperglycemia in type 1 (T1DM) and type 2 diabetic (T2DM) individuals markedly elevates plasma and urinary levels of MGO as a consequence of glycolytic overload. The abnormal accumulation of MGO (dicarbonyl stress) has been implicated in many diseases. Methylglyoxal promotes post-translational modification of peptides or proteins, leading to the formation of advanced glycation end products (AGEs), including hydroimidazolone derived from arginine (MG-H1) and carboxyethyl-lysine (CEL). AGEs bind to their receptor ligand (termed RAGE) anchored in cell membranes, triggering multiple intracellular signaling pathways, leading to increased reactive oxygen species (ROS) production. Under healthy conditions, glyoxalases (Glo) are the most important enzymatic detoxification system converting MGO into its final product D-lactate. Glyoxalases comprise two main enzymes, namely, Glo1 (lactoylglutathione methylglyoxal lyase) and Glo2 (hydroxyacylglutathione hydrolase), with Glo1 described as a rate-limiting enzyme in detoxification. This image was produced with the assistance of Servier Medical Art (Servier; https://smart.servier.com/ accessed on 18 March 2024), licensed under a Creative Commons Attribution 4.0 Unported License.

**Figure 2 biomedicines-12-00939-f002:**
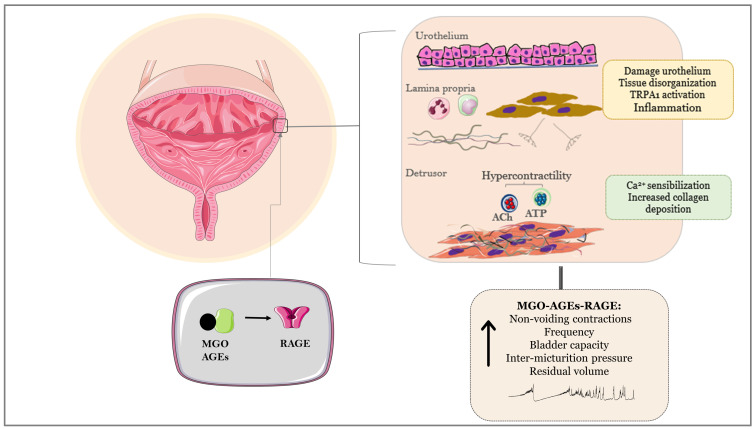
Bladder alterations at the level of urothelium, lamina propria, and detrusor smooth muscle in mice treated with methylglyoxal (MGO) for 4 and 12 weeks. Activation of the MGO–AGEs–RAGE axis leads to urothelial damage, tissue disorganization, edema, and inflammatory cellular infiltration, along with sensitivity alterations due to TRPA1 channel activation. The in vitro detrusor contractile responses to electrical-field stimulation (EFS), α,β-methylene ATP (purinergic P2X1 receptor agonist), and carbachol (nonselective muscarinic agonist) due to increased Ca^2+^ sensitization machinery are higher in MGO-treated mice. Higher collagen deposition is seen in bladders of MGO-treated mice. Urodynamic changes, including increases in nonvoiding contractions (NVCs), frequency, bladder capacity, inter-micturition pressure, and residual volume, may also be observed in MGO groups. Drugs capable of scavenging MGO and protecting bladder cells from oxidative insult, such as the polyphenols resveratrol and epigallocatechin-3-gallate, and the antihyperglycemic metformin exert reduce AGEs levels and oxidative stress in bladder tissues. This image was produced with the assistance of Servier Medical Art (Servier; https://smart.servier.com/ accessed on 18 March 2024), licensed under a Creative Commons Attribution 4.0 Unported License.

**Figure 3 biomedicines-12-00939-f003:**
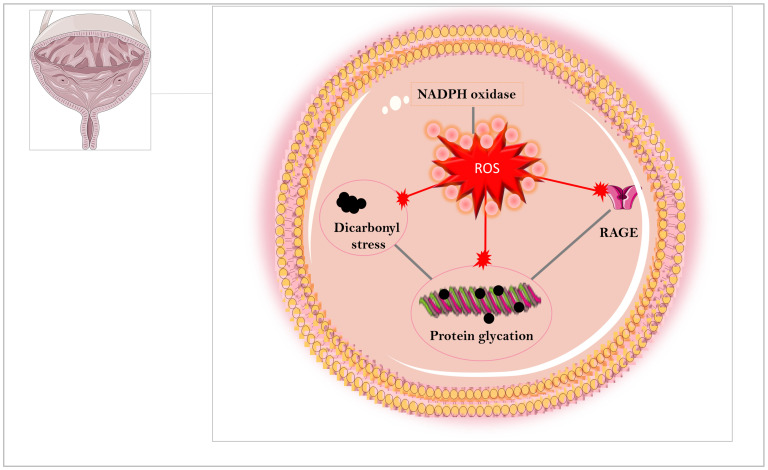
The abnormal accumulation of methylglyoxal (MGO) enhances the dicarbonyl stress leading to protein glycation and excessive RAGE-mediated ROS production in the urinary bladder. This image was produced with the assistance of Servier Medical Art (Servier; https://smart.servier.com/ accessed on 18 March 2024), licensed under a Creative Commons Attribution 4.0 Unported License.

**Table 1 biomedicines-12-00939-t001:** Main findings produced by methylglyoxal (MGO) treatment in rodents.

Reference Number	Dose	Route of Administration	Animal and Strain	Sex	Treatment with MGO
[164]	50–75 mg/kg	Intraperitoneal	Wistar rat	Male	Microvascular damage Microvessel degeneration
[165]	50–75 mg/Kg	Drinking water	Spontaneously diabetic (GK) rats	Male	Endothelial dysfunction
[166]	50 mmol/L	Drinking water	C57Bl6 ApoE^-/-^	Male	Atherosclerosis
[170]	50–75 mg/kg	Drinking water	Goto-Kakizaki (GK), nonobese type 2 diabetic rats	ND	Renal disease
[171]	17.25 mg/kg	Intraperitoneal	Sprague Dawley (SD) rats	ND	Renal disease
[172]	600 mg/kg/day	Oral	NMRI mice	Male	Diabetic nephropathy
[176]	500–2000 mg/kg	Drinking water	RAGE−/Glo1 ++ mice	Male Female	Renal dysfunction
[178]	500 mg/kg	Drinking water	RAGE-KO mice	Male Female	Increased airway resistance/decreased maximal inspiratory flow
[173,175]	0.5%	Drinking water	C57BL/6Junib mice	Male	Aggravation of allergic airway disease and acute lung injury

ND, nondetermined; T2DM, type 2 diabetes mellitus; KO, knockout.

**Table 2 biomedicines-12-00939-t002:** In vivo and in vitro bladder alterations in male and female mice treated with methylglyoxal (MGO) 12 weeks [15,177,180].

	Parameter	Male	Female
Urodynamic evaluation	Number of nonvoiding contractions (NVCs)		
Frequency of voiding	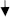	
Bladder capacity		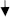
Bladder smooth Muscle contractility in vitro(presence of urothelium)	Neurogenic contractions (electrical-Field Stimulation, EFS)		
Muscarinic-mediated contractions (carbachol)		
Purinergic-mediated contractions (α,β-methylene ATP)		
Void spot analysis	Total void volume		
Volume per void		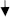
Urine spot number		
Urine spot in center		
Urine spot in corner		
Histology	Collagen content		

Arrows indicate 

 increased; 
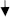
 decreased or 

 unaltered parameters.

**Table 3 biomedicines-12-00939-t003:** Protective effects of alagebrium (ALT-711) on the levels of total AGEs, MG-H1, RAGE and collagen in bladder tissues of obese diabetic ob/ob mice [16].

Parameter	ob/ob	ob/ob + ALT-711
Blood glucose		
AGEs in bladder		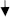
MG-H1 content in bladder		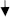
RAGE content in bladder		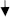
Collagen content in the bladders		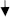
Volume per void		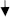
Number of voids	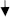	
Void size		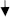

Arrows indicate 

 increased; 
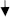
 decreased or 

 unaltered parameters.

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
