# Peer review of "Methylglyoxal and Advanced Glycation End Products (AGEs): Targets for the Prevention and Treatment of Diabetes-Associated Bladder Dysfunction?"

_biomedicines, 2024, doi:10.3390/biomedicines12050939_

Round 1
Reviewer 1 Report
Comments and Suggestions for Authors
1. In lane 49 expand the aberration of OAB
2. There is no references for sentences in introduction
3. Present the graphical representation of Methylglyoxal – Advanced Glycation End Products (AGEs) – RAGE Signaling Pathway
4. No need to present the one study in Table 1 and Table, already described in the paragraphs
Comments on the Quality of English Language
1. This review paper needs some grammatical editing
2. There are typo errors in several places
Author Response
First of all, we would like to sincerely thank the Reviewers #1, #2, #3 and #4 for their good appreciation, critical comments, and great help toward strengthening this review. We agreed with most of the Reviewers´ comments and attempted to modify text according to their main suggestions. We also thank the Editor for allowing us to submit a revised copy of the review. Our point-by-point responses have been included in the Biomedicines system. We have also done some additional editing to improve clarity and language flow.
Reviewer 1
- In lane 49 expand the aberration of OAB
Reply: Please, note that OAB symptoms and causes are relatively well explored a bit further ahead in the text, specifically at item 2.0 (Lower Urinary Tract Symptoms (LUTS) and Overactive Bladder (OAB) Syndrome; Pages 2 and 3).
- There is no references for sentences in introduction
Reply: References have been included in the Introduction of this revised version, as suggested by the Reviewer (Pages 1 and 2, Item 1; refs 1 to 20).
- Present the graphical representation of Methylglyoxal – Advanced Glycation End Products (AGEs) – RAGE Signaling Pathway
Reply: Please, note that Figure 1 seems to fulfill the idea of graphically representing the MGO- AGEs – RAGE signaling. Following this Reviewer´ comment, we moved Figure 1 from item 1 (Introduction, Page 2) to Item 5 (Methylglyoxal – Advanced Glycation End Products (AGEs) – RAGE Signaling
Pathway, Page 6, 2nd paragraph).
- No need to present the one study in Table 1 and Table, already described in the paragraphs
Reply: We think tables 1 and 2 should be kept in the review, provided they reinforce the findings with long-term intake of MGO on in vitro and in vivo bladder alterations, which is lacking in literature. Please, also note that Reviewer #2 asked us to redesign the tables, and Reviewer #4 even suggested us to improve both tables, making them colorful and more original.
Reviewer 2 Report
Comments and Suggestions for Authors
The authors propose intriguing mechanisms that could be biomarkers and/or drug targets in diabetic bladder dysfunction.
Conceptual comments:
1. L. 18-20, 47-50, 90-95 and elsewhere: While it is true that urgency and detrusor overactivity can occur in diabetic patients (particularly in early phases of the condition), detrusor underactivity can also occur (particularly in later phases of the condition). Therefore, the sole focus on OAB/urgency incontinence here is too limited. It would be fine to focus solely on overactivity, but that should be stated clearly if you wish to make this choice. Of note, symptoms such as incontinence can result both from detrusor over- and underactivity (the latter mostly being overflow incontinence), and the differentiation between the two can be challenging in patients. Of note, overflow incontinence is mentioned in l. 134 and 159 but should also be more generally be mentioned.
2. The big question about MGO or AGEs being a possible drug target is whether they indeed are mediators or “only” biomarkers. Most of the evidence being presented here would support a biomarker status. However, at least for bladder, the evidence for cause effect relationships for dysfunction appears much weaker. While I explicitly do not wish to rule the hypothesis of the authors, I am skeptical about how strong the evidence is (see also l. 300). While the strongest available evidence appears to come from the MGO treatment studies, I wonder whether MGO administration mimics only the overactive or also the underactive phenotype of diabetes-associated bladder dysfunction. For instance, Table 1 suggests that in male mice 4 weeks of MGO increases voiding frequency whereas 12 weeks decreases it. The manuscript may get scientifically stronger if this uncertainty is discussed transparently.
3. If MGO works in bladder dysfunction, does it do so by improving glucose homeostasis or by acting directly on the bladder? If the former, why is it more than any other type of anti-diabetic compound? If the latter, why restrict it to diabetic cystopathy?
4. Given that Table 1 demonstrates that the effects of MGO administration differ markedly between male and female mice, I wonder whether the sex/gender of animals and patients being studied needs to be systematically mentioned for all studies throughout the manuscript. If we trust Table 1, sex can be a key factor here. Surprisingly, the related text (L. 318-321) does not talk about these major sex differences. Any speculation why MGO administration for 12 weeks works so differently in male and female mice? If these findings are true, would drugs targeting MGO/AGE be likely to work in only one gender?
Other comments:
5. As all abbreviations are reintroduced in the main paper, does it make sense to introduce the abbreviations MG-H1, CML and ALT-711 in the Abstract? They are not used any further here. Interestingly, the abbreviation MG-H1 is used in the main paper whereas CML is not (l. 249).
6. L. 37, 152, 234, 238, 304, 307, 315 and elsewhere: “significantly” is ambiguous here because readers cannot understand whether the plain English meaning (i.e., important or similar) or the statistical meaning (i.e., p-value smaller than the statistical alpha) is meant; of note, the statistical meaning does not always associate with the plain English meaning, i.e., an effect can be of putative relevant size but have a high p-value or the opposite. Therefore, many statisticians recommend replacing “significant” by what is meant in a context.
7. L. 50: Is a period missing prior to „Increased“ or is the capital I a typo?
8. L. 52-53, 173-174, 444 and elsewhere: While a high fructose diet can (but not consistently does) lead increased glucose levels, it is mostly used as a model of NASH or similar types of liver disease. The general diabetes community does not recognize it as a diabetes value of translational value. Can you replace by a T2DM model that is more representative, e.g., db/db mice or obese Zucker or STD rats? Moreover, a high-fat diet unless accompanied by additional intervention such as low-dose STZ typically does not lead to diabetes.
9. L. 35-72 provides important general background information but the entire paragraph is devoid of suitable references. Given the scope of this paragraph, I would have assumed that it requires at least a dozen references.
10. L. 99-100: According to the ICS definition, urgency is the only mandatory symptom of OAB whereas all others are facultative (urgency incontinence occurring in only about one third of OAB patients in epidemiological surveys).
11. L. 102-104: I find this claim to be factually wrong. The most frequent LUTS in women is stress urinary incontinence. Please reword.
12. L. 106-109: This reference is about LUTS in general and in no way specific to OAB, BOO, BPH, which are the topics of the rest of the paragraph. This difference should be stated more clearly. For instance, epidemiological studies applying the ICS definition of OAB across multiple countries found a prevalence of 11-13%, which is much lower than the 31.3% or 63.2% stated here.
13. Section 3: While I agree with the general conclusion of this section, it should be noted that not all individual studies in this field have detected an association between metabolic syndrome and bladder dysfunction. Moreover, the improved incontinence upon weight loss is largely explained by less stress urinary incontinence. Please reword.
14. L. 18, 129 and elsewhere: While I agree that bladder dysfunction may be the most frequent complication of diabetes, I am less certain about “most bothersome”. Wouldn’t losing eye sight, getting a foot amputated, or having to undergo dialysis treatment be more bothersome? Please consider rewording.
15. L. 147-148: While this is correct for insulin, reference #62 reports that this is not generally true for treatments typically applied in T2DM. Please check to avoid misinterpretation.
16. L. 231-232: Are they normalized by transgenic expression of Glo-1 (implying expression after diabetes) or prevented (implying expression prior to diabetes)?
17. L. 233-245: Given the huge variability of plasma MGO in healthy subjects (0.21-400 µM), it is not informative to say that they are elevated in diabetes. Can you give readers an impression on the degree of increase? Same for urinary MGO.
18. Table 1: It would be helpful to mention the reference on which the table is based in the table legend. Moreover, it may be helpful to design the table in a way that direct comparison between data at 4 and 12 weeks can be assessed more easily.
Author Response
First of all, we would like to sincerely thank the Reviewers #1, #2, #3 and #4 for their good appreciation, critical comments, and great help toward strengthening this review. We agreed with most of the Reviewers´ comments and attempted to modify text according to their main suggestions. We also thank the Editor for allowing us to submit a revised copy of the review. Our point-by-point responses have been included in the Biomedicines system. We have also done some additional editing to improve clarity and language flow.
- L. 18-20, 47-50, 90-95 and elsewhere: While it is true that urgency and detrusor overactivity can occur in diabetic patients (particularly in early phases of the condition), detrusor underactivity can also occur (particularly in later phases of the condition). Therefore, the sole focus on OAB/urgency incontinence here is too limited. It would be fine to focus solely on overactivity, but that should be stated clearly if you wish to make this choice. Of note, symptoms such as incontinence can result both from detrusor over- and underactivity (the latter mostly being overflow incontinence), and the differentiation between the two can be challenging in patients. Of note, overflow incontinence is mentioned in l. 134 and 159 but should also be more generally be mentioned.
Reply: In our hands, using the models of prolonged MGO intake (up to 12 weeks) to healthy normoglycemic mice, we have never seen in vitro detrusor underactivity or in vivo alterations entirely compatible with bladder underactivity. Therefore, we focused this review mostly on bladder/detrusor overactivity phenotype. In diabetic ob/ob mice at 4-month-old age we do not clearly see detrusor underactivity either [16], whereas at 6-month-old (unpublished data), these animals exhibit some degree of detrusor underactivity (reduced in vitro contraction) associated with in vivo high capacity; and the spot volume is still high. Anyhow, following the Reviewer´s suggestion, we expanded the information on detrusor underactivity at the Introduction (Page 2, lines 52-60). Note also that some information on detrusor underactivity had already been described when mentioning the streptozotocin (STZ) and Akita T1DM models (please, see Item 4, Bladder Dysfunction in Type 1 and Type 2 Diabetes in Patients and Animals; Page 4, lines 149-156). In Introduction of this revised version, we stated that our review focused on “bladder overactivity”, as suggested (Page 2, lines 82-83). We also rephrased some pieces here attempting to avoid the term “overflow incontinence” because we realized it is more appropriate when describing the symptoms / pathophysiology.
The big question about MGO or AGEs being a possible drug target is whether they indeed are mediators or “only” biomarkers. Most of the evidence being presented here would support a biomarker status. However, at least for bladder, the evidence for cause effect relationships for dysfunction appears much weaker. While I explicitly do not wish to rule the hypothesis of the authors, I am skeptical about how strong the evidence is (see also l. 300). While the strongest available evidence appears to come from the MGO treatment studies, I wonder whether MGO administration mimics only the overactive or also the underactive phenotype of diabetes-associated bladder dysfunction. For instance, Table 1 suggests that in male mice 4 weeks of MGO increases voiding frequency whereas 12 weeks decreases it. The manuscript may get scientifically stronger if this uncertainty is discussed transparently.
Reply: In our hands, prolonged MGO treatment up to 12 weeks just produces an overactivity pattern, but we understand we should extent this treatment for prolonged periods (e.g., 24 or 48 weeks) in order to find out if underactivity appears. When data of male mice from 4- and 12-week MGO treatment are compared (original Table 1), the voiding frequency indeed seems to decline as treatment progresses, but even at 12 weeks we still see in vitro detrusor overactivity. We perhaps should move to more prolonged times of treatment with MGO (say, 24 or 48 weeks of MGO treatment) to see if underactivity turns up. Regarding the biomarkers or mediators’ status of the MGO-AGEs-RAGE axis, this is a challenging question, the answer of which is open to discussion. We certainly believe that MGO, AGEs and RAGE levels in bladder tissues are good biomarkers of bladder dysfunction in obese / diabetic mice, as we have emphasized in this review. Concerning the possibility of each component of this pathway also acting as “mediators”, we have evidence that could be the case. For instance, in ob/ob mice, the increased levels of MGO (serum), total AGEs, MG-H1 and RAGE expression (bladder tissues) were all reversed by the AGEs breaker alagebrium, and that was accompanied by reductions of the volume per void and the number of spots [16]. In healthy normoglycemic mice treated with MGO for 12 weeks, treatment with anti-hyperglycemic drug metformin normalized the MGO levels in serum (possibly by scavenging MGO in blood / tissues) that resulted in normalization of the bladder dysfunction [177]. Therefore, although we understand that different experimental strategies using other AGEs / RAGE inhibitors and models of obesity / diabetes are needed before a conclusion can be made, we believe that MGO, AGEs and RAGE behave both as mediators and biomarkers.
- If MGO works in bladder dysfunction, does it do so by improving glucose homeostasis or by acting directly on the bladder? If the former, why is it more than any other type of anti-diabetic compound? If the latter, why restrict it to diabetic cystopathy?
Reply: As mentioned in item 6 (3rd paragraph), the long-term intake of MGO to normoglycemic (non-diabetic) mice change neither the fasting glucose levels nor the insulin sensitivity [173,177], strongly suggesting that bladder dysfunction in such animals reflect directly the MGO accumulation and excess AGEs formation and RAGE activation in the bladder tissues. We have also planned to work with genetically diabetic ob/ob mice (which according to our experience they exhibit high endogenous levels of MGO) further supplemented with exogenous MGO in the drinking water to see if hyperglycemia and insulin sensitivity further increases, worsening the bladder dysfunction (or making the overactivity progressing to underactivity like the T2DM non-controlled individuals). We are not sure if the pharmacological inhibition of the components of the MGO-AGEs-RAGE axis to treat diabetic cystopathy would have the same efficacy of any other commercially available anti-diabetic compound that normalizes the glucose levels (and hence theoretically the MGO levels) in the diabetic individuals. The protein glycation by dicarbonyl species like MGO is said to be a gradual and irreversible process [see ref 199]; therefore, under certain circumstances, drugs able to specifically break the cross-linking (or those that selectively block RAGE) may show advantages over the classical anti-diabetic compounds.
Given that Table 1 demonstrates that the effects of MGO administration differ markedly between male and female mice, I wonder whether the sex/gender of animals and patients being studied needs to be systematically mentioned for all studies throughout the manuscript. If we trust Table 1, sex can be a key factor here. Surprisingly, the related text (L. 318-321) does not talk about these major sex differences. Any speculation why MGO administration for 12 weeks works so differently in male and female mice? If these findings are true, would drugs targeting MGO/AGE be likely to work in only one gender?
Reply: Indeed, here, we did not compare data of male and female mice treated with MGO for 4 or 12 weeks [14,15,177,180]. In part, this was due to the fact that our studies in MGO-treated mice were not carried out simultaneously in male and female animals, but at different times. In addition, in the beginning, when we performed the 4-week MGO treatment, we employed just male mice, allowing no comparison between sex. For the 12-week MGO treatment, we indeed have data for both sexes. Using this treatment duration, our data show voiding differences between male and females, particularly those related the frequency and capacity, as evaluated by the void spotting assay on filter paper (Table 1). However, the in vitro contractile responses (organ bath experiments) generally show a higher response in the MGO groups irrespective of the animal sex, as evidenced using both electrical-field stimulation (EFS; neurogenic response) and the purinergic receptor agonist α,β-methylene ATP. On the other hand, the increased contractile response to muscarinic receptor activation (carbachol) in the female group are just seen when mucosa (urothelium) is removed, whereas in males it occurs in both intact and denuded preparations (we do not have a reasonably good explanation here for this “sex discrepancy” and did not attempt to further explore it in the review). We recently moved to investigate the bladder responses in obese diabetic ob/ob mice, testing simultaneously male and female animals of same ages [16]. According to our experience, ob/ob mice have higher endogenous levels of MGO (plasma), total AGEs, MG-H1 and RAGE (bladder tissues) in comparison with wild-type non-diabetic group. These animals exhibit marked voiding alterations compared with respective sex controls, but we did not see great differences between male and female groups regarding the in vivo voiding pattern and the organ bath experiments. Therefore, we are still uncertain on the emphasis we should give to the apparent discrepancies on the voiding behavior pattern between sex and prefers not to go further in such comparisons before we achieve more data. We hope the Reviewer agrees with our opinion on this.
Other comments:
- As all abbreviations are reintroduced in the main paper, does it make sense to introduce the abbreviations MG-H1, CML and ALT-711 in the Abstract? They are not used any further here. Interestingly, the abbreviation MG-H1 is used in the main paper whereas CML is not (l. 249).
Reply: We have done that according to Biomedicines style for publication, which “spell out full” each abbreviation in the Abstract and at the first time they appear in the Introduction section. Anyhow, we are double-checking this information before resubmitting this revised version.
L. 37, 152, 234, 238, 304, 307, 315 and elsewhere: “significantly” is ambiguous here because readers cannot understand whether the plain English meaning (i.e., important or similar) or the statistical meaning (i.e., p-value smaller than the statistical alpha) is meant; of note, the statistical meaning does not always associate with the plain English meaning, i.e., an effect can be of putative relevant size but have a high p-value or the opposite. Therefore, many statisticians recommend replacing “significant” by what is meant in a context.
Reply: We have backtracked each “significant” / “significantly” in text and changed accordingly. The most “significant / significantly” we employed were indeed due to statistical meaning at p<0.05. Nevertheless, we removed some “significant” and replaced some by “marked” or “markedly” accordingly.
- L. 50: Is a period missing prior to „Increased“ or is the capital I a typo?
Reply: There was a missing “dot” in the phrase, which was now corrected.
L. 52-53, 173-174, 444 and elsewhere: While a high fructose diet can (but not consistently does) lead increased glucose levels, it is mostly used as a model of NASH or similar types of liver disease. The general diabetes community does not recognize it as a diabetes value of translational value. Can you replace by a T2DM model that is more representative, e.g., db/db mice or obese Zucker or STD rats? Moreover, a high-fat diet unless accompanied by additional intervention such as low-dose STZ typically does not lead to diabetes.
Reply: We thank the Reviewer for updating us on the fructose model (we personally have no experience with this model). Following the Reviewer view, we have removed the fructose model from the revised version with its respective references (Lee WC et al., J Urol. 2011,186(1):318-25; and Tong & Cheng, J Urol. 2007, 178(4 Pt 1):1537-42).
L. 35-72 provides important general background information but the entire paragraph is devoid of suitable references. Given the scope of this paragraph, I would have assumed that it requires at least a dozen references.
Reply: We have now inserted key references for the entire text in the Introduction (Pages 1 and 2, Item 1; refs 1 to 20). References here had not been inserted before because we meant this entire whole paragraph as a “preface” that would be expanded while the review was constructed.
- L. 99-100: According to the ICS definition, urgency is the only mandatory symptom of OAB whereas all others are facultative (urgency incontinence occurring in only about one third of OAB patients in epidemiological surveys).
Reply: We have rephrased text to make this information clear (Page 2, Item 2, Lines 89-97; Page 3, Line 98-99).
L. 102-104: I find this claim to be factually wrong. The most frequent LUTS in women is stress urinary incontinence. Please reword.
Reply: We have rephrased, as suggested by the Reviewer (Page 3, Lines 98-99).
L. 106-109: This reference is about LUTS in general and in no way specific to OAB, BOO, BPH, which are the topics of the rest of the paragraph. This difference should be stated more clearly. For instance, epidemiological studies applying the ICS definition of OAB across multiple countries found a prevalence of 11-13%, which is much lower than the 31.3% or 63.2% stated here.
Reply: We have decided to remove the information related to “Huang J et al. Prostate Cancer Prostatic Dis. 2023, 26(2):421-428” and have added the information on ICS definition (Page 3, Lines 102-104), as suggested by the Reviewer.
Section 3: While I agree with the general conclusion of this section, it should be noted that not all individual studies in this field have detected an association between metabolic syndrome and bladder dysfunction. Moreover, the improved incontinence upon weight loss is largely explained by less stress urinary incontinence. Please reword.
Reply: We have included two new references [45,46] that show no relationship between metabolic syndrome and LUTS (Page 03, Item 3, Lines 113-114). We also rewrote the text to make clear that the benefits of weight loss are more restricted to the stress urinary incontinence (Page 03, Lines 118,119).
- L. 18, 129 and elsewhere: While I agree that bladder dysfunction may be the most frequent complication of diabetes, I am less certain about “most bothersome”. Wouldn’t losing eye sight, getting a foot amputated, or having to undergo dialysis treatment be more bothersome? Please consider rewording.
Reply: We removed “the most”, leaving just “a bothersome”.
- L. 147-148: While this is correct for insulin, reference #62 reports that this is not generally true for treatments typically applied in T2DM. Please check to avoid misinterpretation.
Reply: Here, we are not sure we understood exactly what checking the Reviewer wanted us to do here. In ref #62 (now ref 80) we just referred to mechanisms involved in bladder enlargement, whereas in refs #63-66 (now refs 79,81,82,83) we referred to insulin administration in T1DM (STZ) model.
L. 231-232: Are they normalized by transgenic expression of Glo-1 (implying expression after diabetes) or prevented (implying expression prior to diabetes)?
Reply: In ref #122 (now ref 123), it is stated that in transgenic GLO-1 rats (prior to diabetes), intravenous injection of STZ (65 mg/kg) was used to induce diabetes. We changed a little bit the text to make this clearer (Page 5, Lines 224,225).
- L. 233-245: Given the huge variability of plasma MGO in healthy subjects (0.21-400 µM), it is not informative to say that they are elevated in diabetes. Can you give readers an impression on the degree of increase? Same for urinary MGO.
Reply: This great variability possibly reflects the analytical methods employed to quantify MGO in plasma / serum, which fairly differs across studies and often lacks details for those who have not expertise (such as if extractions were done previously or not). The studies quoted in this paragraph dealt with diabetic patients (defined by the guidelines or Diabetes Associations of each respective country) presenting different characteristics, as follows: men and women at variable ages (54 to 61 y.o.) with accompanying chronic renal failure, diabetic nephropathy, or coronary heart disease; insulin-users and non-insulin users. Following the Reviewer ‘comments here, we excluded from the revised version the information on the concentration range of MGO (0.21-400 µM) and inserted a short paragraph pointing out the characteristics of the patients above-mentioned (Page 6, Lines 260-262).
- Table 1: It would be helpful to mention the reference on which the table is based in the table legend. Moreover, it may be helpful to design the table in a way that direct comparison between data at 4 and 12 weeks can be assessed more easily.
Reply: We have included the references in tables 1 and 2 (now referred as table 2 and 3). We have attempted to change tables according to the Reviewers #2 and #4 comments.
Reviewer 3 Report
Comments and Suggestions for Authors
THe review is quite comprehensive. THe references are up to date.
All the data presented pertains to in vitro and animal studies.
No clinical data is available.
he authors need to add in the manuscript under summary that there are no clinical trials that looked at methods of inhibitng Methylglyoxal and Advanced Glycation End Products (AGEs) and their ability to reliev the manifestations of diabetes induced/ Associated Bladder Dysfunction.
Comments on the Quality of English Languageok
Author Response
First of all, we would like to sincerely thank the Reviewers #1, #2, #3 and #4 for their good appreciation, critical comments, and great help toward strengthening this review. We agreed with most of the Reviewers´ comments and attempted to modify text according to their main suggestions. We also thank the Editor for allowing us to submit a revised copy of the review. Our point-by-point responses have been included in the Biomedicines system. We have also done some additional editing to improve clarity and language flow.
The authors need to add in the manuscript under summary that there are no clinical trials that looked at methods of inhibitng Methylglyoxal and Advanced Glycation End Products (AGEs) and their ability to reliev the manifestations of diabetes induced/ Associated Bladder Dysfunction.
Reply: We have added the above suggested phrase with a slight modification (Please, see Abstract, Lines 28-30; and Page 2, Lines 75-77).
Reviewer 4 Report
Comments and Suggestions for Authors
The authors based on methylglyoxal, which accumulates in the plasma and urine of individuals with glycemia and diabetes (as a potent peptide glycating molecule) causes a preaccumulation of advanced glycation end products (AGEs), oxygen species (ROS), kakato and diabetic dysfunction and accumulation in bladder tissues. The authors postulate that MGO, AGE, and RAGE may be critically involved in the pathogenesis of bladder dysfunction in diabetic patients.
Minor comments:
1. row 47-50 to be reviced
2. row 60-64 to be reviced
3.row 104-107 to be reviced
4. T1DM mimicked by streptozotocin (STZ) injection in rodents schematically;
5.The leptin model is presented schematically
6.row 234-240 to paraphrase
7. in rows 285-330 many abbreviations were used
8.chronic overload intake of MGO at doses of 50 to 75 mg/k....... to be presented by cheme
9.table 1 and 2 are boring. to present themselves in a colorful and more original way
10.superoxide an- 363 ion and hydrogen peroxide, to be present by ionic structures
11. to be add a scheme for oxidative stress; to be add more references
12. the metformin use is incomplete
13. the references from the last 3 years are only 11 %.
Comments on the Quality of English Language
Minor editing of English language required
Author Response
First of all, we would like to sincerely thank the Reviewers #1, #2, #3 and #4 for their good appreciation, critical comments, and great help toward strengthening this review. We agreed with most of the Reviewers´ comments and attempted to modify text according to their main suggestions. We also thank the Editor for allowing us to submit a revised copy of the review. Our point-by-point responses have been included in the Biomedicines system. We have also done some additional editing to improve clarity and language flow.
The authors based on methylglyoxal, which accumulates in the plasma and urine of individuals with glycemia and diabetes (as a potent peptide glycating molecule) causes a preaccumulation of advanced glycation end products (AGEs), oxygen species (ROS), kakato and diabetic dysfunction and accumulation in bladder tissues. The authors postulate that MGO, AGE, and RAGE may be critically involved in the pathogenesis of bladder dysfunction in diabetic patients.
Minor comments:
- row 47-50 to be reviced
Reply: There was a missing “dot” in the phrase, which was now corrected.
- row 60-64 to be reviced
Reply: We have rewritten this paragraph, as suggested by the Reviewer (Page 2, Lines 69-74).
3.row 104-107 to be reviced
Reply: We have rewritten the paragraph, as suggested by this Reviewer #4 (and Reviewer #2; Page 3, Lines 99-102).
T1DM mimicked by streptozotocin (STZ) injection in rodents schematically; &
5.The leptin model is presented schematically
Reply: We are not sure we correctly understood the Reviewer suggestion here at items 4 and 5. Nevertheless, the STZ and leptin model are widely have been known for a long time. We think a scheme here would not help that much the review.
6.row 234-240 to paraphrase
Reply: We have modified the paragraph and inserted a new sentence, according to the Reviewer´s suggestions (Page 6, Lines 257-262). Please, note that Reviewer #2 also suggested modifications here.
in rows 285-330 many abbreviations were used
Reply: Indeed, there was no need to keep “SDT” abbreviation (referring to spontaneously diabetic Torii) in the text, and so we removed it. The other abbreviations comprising this part (T2DM, LUTS, AGEs, MG-H1, MGO, NVCs, EFS and TRPA1; Lines 296 to 360) were maintained because it would complicate spelling out full them at this point of the review.
8.chronic overload intake of MGO at doses of 50 to 75 mg/k….... to be presented by scheme
Reply: We appreciated the Reviewer´s suggestion and inserted a new table (Table 1 of the revised version), describing data so far obtained using MGO treatments in mice and rats. Specifically, we detailed the MGO doses, routes of administration, animal species and strain, and main findings.
9.table 1 and 2 are boring. to present themselves in a colorful and more original way
Reply: Table #1 has been redrawn, attempting to make it more illustrative, as suggested by the Reviewer.
10.superoxide and ion and hydrogen peroxide, to be present by ionic structures
Reply: Superoxide and hydrogen peroxide were also presented as ionic structures at the first time they appear, as suggested (Page 11, Line 418,419).
to be add a scheme for oxidative stress; to be add more references
Reply: We appreciated the Reviewer´s suggestion and inserted a new figure (Figure 3 of the revised version) containing an scheme of NADPH oxidase / ROS amd its relationships with protein glycation and RAGE.
- the metformin use is incomplete
Reply: We expanded the metformin use, according to the Reviewer´s suggestion (Page 12, Item 7.2, Lines 481-483).
- the references from the last 3 years are only 11 %.
Reply: If we consider the last 5 years, it reaches nearly 40%. In contrast with other pathological conditions like cardiovascular diseases, there are indeed not many publications exploring the role of the MGO-AGEs-RAGE signaling on obesity- and diabetes-associated bladder dysfunction, that is why the present review may make a difference in the literature.
Round 2
Reviewer 2 Report
Comments and Suggestions for Authors
All of my comments have been addressed adequately.
Nice paper, as expected from the Antunes group.
Reviewer 4 Report
Comments and Suggestions for Authors
-